# Seasonal and Altitudinal Variation in Chemical Composition of *Celtis australis* L. Tree Foliage

**Bhupendra Singh [1,2]**, **Munesh Kumar [1,*]**, **Marina M. S. Cabral-Pinto [3,*]** and **Bhagwati Prasad Bhatt [4]**

1  Department of Forestry and Natural Resources, H.N.B. Garhwal University (A Central University), Srinagar Garhwal 249161, Uttarakhand, India
2  College of Forestry, V.C.S.G. Uttarakhand University of Horticulture and Forestry, Ranichauri, Tehri Garhwal 249199, Uttarakhand, India
3  GeoBioTec Research Centre, Department of Geosciences, University of Aveiro, 3810-193 Aveiro, Portugal
4  Natural Resource Management Division, ICAR, KAB-II, Pusa, New Delhi 110012, India
*  Correspondence: muneshmzu@yahoo.com (M.K.); marinacp@ua.pt (M.M.S.C.-P.)

**Abstract:** *Celtis australis* is an important agroforestry tree in the Himalayan region providing major fodder to the livestock and many other needs for sustaining life in local rural communities. Including their fodder value and wide altitudinal distribution ranges, it is extracted by the villagers in large scale during the lean period (May to June). Thus, the aim was to understand the nutritive values of the species influenced by the altitude and harvesting season. For this investigation, leaves of *C. australis* were collected from four different altitudes during the months of February to December. The nutritive value of collected foliage, i.e., dry matter, ash, potassium, calcium, phosphorus, crude protein, crude fiber, starch, sugar, and phenolic were estimated by using stranded methods. The results of present study reveal that significant seasonal variations have been observed for the chemical composition of *C. australis* tree foliage collected from different altitudes. On an average, between seasons, crude protein ranged from 9.17 to 16.97%, phosphorus 0.08 to 0.16%, potassium 0.28 to 0.76%, crude fiber 13.94 to 19.80%, sugars 1.11 to 1.83%, and starch 4.79 to 6.53%. Altitude also significantly influenced nutritive content. Between altitudes, phosphorus ranged from 0.10 to 0.11%, potassium 0.42 to 0.50%, crude protein 12.66 to 14.02%, crude fiber 1.61 to 1.71%, sugars 1.45 to 1.66%, starch 4.71 to 6.31%, and phenolic 0.43 to 0.78%. Potassium, calcium, crude protein, and starch were significantly positively correlated with altitude of foliage collection. However, dry matter content, phosphorus, and soluble sugars, significantly correlated inversely with altitude.

**Keywords:** *Celtis australis*; altitude; nutritive value; composition change

## 1. Introduction

Fodder trees are convenient for providing various nutrients to the livestock population in hilly regions. In mountain villages of India, along an elevation gradient, as many as 42 multipurpose trees and shrubs have been cultivated in or around agricultural fields in different agroforestry systems for fodder, fuel, fiber, timber, and various other miscellaneous uses [1]. Livestock of the Himalayan region of India depends on trees and shrubs for fodder during different seasons, where the fodder tree species change from lower to higher altitudes. *Celtis australis* belongs to the family Cannabaceae (earlier Ulmaceae), locally known as Kharik and the common name is Nettle tree. *C. australis* is a moderate-sized, deciduous tree, attaining a height of 25–30 m and a diameter of 60–80 cm. The stem is straight, cylindrical, smooth, and exhibits bluish-gray bark [1–3]. *C. australis* grows in the Western Himalayas eastward to Nepal. It is distributed extensively in Manipur and commonly cultivated in Jummu and Kashmir, Himachal Pradesh, and Uttarankhand [1–3].

It grows along stream banks, on sloping hillsides, and in shallow soils. It is frost hardy and moderate light-demanding, but seedlings and saplings can withstand moderate shade. Usually the species used for fuel, fodder, and small timber [4]; fruits of *C. australis* are eaten

by birds, monkeys, rats, and rodents, and the tree is generally propagated by seeds and branch cuttings [5–8]. Its wood is tough, strong, and elastic, and used for oars, tool handles, sports goods, and fuelwood (with 464 fuelwood index), which contains 16.81 kJ/g calorific value, 0.54 g/cc density, 3.4% ash, 57.53% moisture, and 0.40% nitrogen [9]. It is considered one of the most preferred fodder trees in the Central Himalayan region and harvested by local farmers in the months of April and May when there is usually a deficiency of green fodder. Over-exploitation of *C. australis* has been reported at different altitudes in the Central Himalaya [10]. Owing to the multipurpose value of *C. australis*, it is mainly grown in agrisilviculture and silvopastoral agroforestry systems throughout the hills. It is an agroforestry species that has wide ranging ecological amplitude (extending from 500 to 2500 masl) and, being a fast-growing species, helps to sequester atmospheric carbon and enhance the resilience of climatic variability. Tree species improve the microclimate conditions of the growing region and contribute to climate variability mitigation and adaptation [11–13].

*C. australis* leaves are very broad and highly palatable. Leaves are free from tannin during its peak production [14,15] and play a major role in meeting the nutrient demands within the hilly regions. Animal husbandry is a keystone of Himalayan farming systems and almost all families rear livestock, which depends upon the socioeconomic status of the family. The livestock generate farmyard manure, which is the only source of agricultural crop production, including milk production for the sustenance of the poor farmer community in this part of the Himalaya.

Green fodder (tree/shrub) harvested from forests and agroforestry together with dry fodder (crop byproducts) are the main sources of the diet for the livestock [10]. However, due to a substantial increase in the human and livestock population and decreasing forest areas during the last few decades, the local inhabitants are facing acute shortages of fuel, fodder, and timber resources. The existing resource scarcity, particularly fodder, could be mitigated by judicious use of fodder trees.

Since *C. australis* is multifarious tree-crop of the region, therefore, an attempt has been made to evaluate its fodder quality as influenced by season and altitude. The objective was to screen the suitable populations (those altitudinal populations, which possess high nutritive value) and to record the best season/months when the foliage has peak values for most of the nutritive parameters, because the chemical composition of the leaves and their digestibility, and the species of ruminants that can make use of the tree fodder, are the factors that determine the consumption pattern [16]. Therefore, the nutrient composition of tree leaves at different times during growth can be used as an indicator to determine the appropriate lopping period for this species. In the Garhwal Himalaya, fodder from *C. australis* is harvested twice a year (May–June and October–November). The systematic study of *C. australis* tree fodders in relation to chemical composition is scanty; hence, the hypothesis of the present study was that the nutrient composition of *C. australis* foliage changes with altitude and season. To understand the importance of this promising agroforestry tree crop, particularly in the Garhwal Himalaya, and fulfill the hypothesis, the objective of the study was to determine the change in chemical composition of foliage in different seasons and at different altitudes in the Garhwal Himalaya.

## 2. Materials and Methods

### 2.1. Study Site Characteristics

The study was conducted at H.N.B. Garhwal University Srinagar (Garhwal), Uttarakhand, India. Generally, the land-use types in the Garhwal Himalaya are forests, agroforestry, uncultivated, wasteland, etc. The average annual precipitation varies from 1200 to 2500 mm year$^{-1}$ and the maximum temperature exceeds 38 °C, while the minimum is as low as $-4$ °C [3]. *C. australis* grows on a variety of soils, preferring deep loamy silts and clays, and favoring stream banks, sloping hillsides, and gravelly shallow soils [3], but cannot withstand impeded drainage and stunted growth occurs in dry gravelly shallow soils. Soil pH, usually under agroforestry systems, varies from 6.20 to 6.44, water-holding

capacity ranges from 48.90 to 52.59%, soil moisture from 8.34 to 15.36%, and soil organic carbon ranges between 1.61 to 1.73% [17]. The soil nutrient status under *C. australis* was also reported by Panwar and Gupta [18], i.e., pH (6.52), EC (mS) (0.17), density (1.14%), soil organic carbon (1.20%), organic matter (2.07%), available nitrogen (0.74%), available phosphorus (0.089%), and exchangeable potassium (0.64%).

The traditional agroforestry system of the Garhwal Himalaya (Uttarakhand) is classified into various categories, i.e., agrosilviculture, agrohorticulture, and agrosilvohorticulture, and farmers cultivate different seasonal, biennial, and perennial crops, i.e., *Echinochloa frumentaceae*, *Eleusine coracana*, *Triticum aestivum*, *Amaranthus* spp., *Vigna umbellata*, and *Oryza sativa*, etc. Other associated species reported in these systems are *Grewia optiva*, *Quercus leucotrichophora*, *Prunus cerasoides*, *Pyrus pasha*, *Ficus* spp., etc., in the upper altitudes, and *Toona ciliata*, *Melia azadirach*, *Bauhinia* spp., *Ficus* spp., *Grewia optiva*, etc., in the lower altitude of agroforestry fields [19]. Seasonally, domestic animals use harvested fields for grazing.

### 2.2. Tree Leaf Samples

The leaves of *C. australis* were collected from four different sites at altitudes ranging from 550 to 1980 masl with geocoordinates 30°06′ to 30°25′ N latitude and 78°38′ to 78°48′ E longitude (Table 1 and Figure 1). Leaves are usually not found in January and February and were therefore collected within a period of 10 months (from March to December) from each site in first week of the month. Five ideotype trees (generally with height ranging from 10 to 12 m, diameter 20–30 cm at breast height, and ages of 20–40 years) were selected for the collection of leaves from each site. From each tree, 400 g of leaves were harvested from all four directions in the lower, middle, and top portions of the canopy, packaged separately in poly bags, and used as replicated. The fresh samples of leaves were sun dried until a constant dry weight was recorded. Dried leaves were crushed in a mechanical grinder to obtain a fine powder that was used to determine their chemical composition.

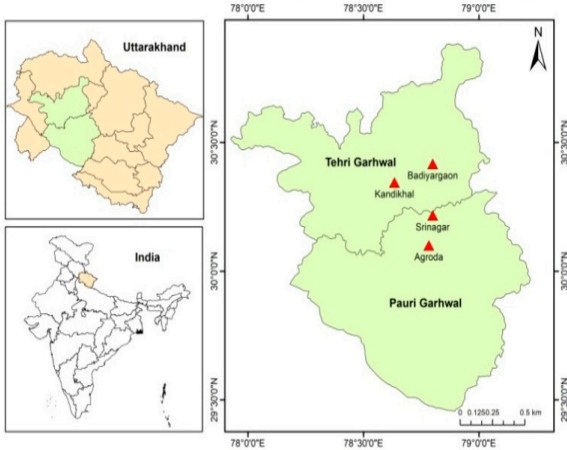

**Figure 1.** Map of the study area location.

**Table 1.** Geographical description of *C. australis* foliage collection sites.

| Provenance | Altitude (masl) | District | Latitude (N) | Longitude (E) | Aspect | Temperature °C | Rainfall (mm·year$^{-1}$) | Soil pH |
|---|---|---|---|---|---|---|---|---|
| Srinagar | 550 | Pauri | 30°13′ | 78°48′ | South | 27.5 | 1145 | 6.08 |
| Agroda | 1180 | Pauri | 30°06′ | 78°47′ | West | 23.0 | 1215 | 6.28 |
| Kandikhal | 1550 | Tehri | 30°20′ | 78°38′ | North | 21.0 | 1270 | 6.14 |
| Badiyargaon | 1980 | Rudraprayag | 30°25′ | 78°48′ | South | 18.0 | 1456 | 6.28 |

### 2.3. Chemical Composition Analysis

Dry matter contents and ash contents were calculated and estimated [20]. The crucibles were dried by placing them in an electric oven at 100 ± 1 °C and cooled in desiccators

before recording their weight. Ten grams (10 g) of fresh leaf material was put into a crucible in an electric oven set at $100 \pm 1$ °C for 24 h and then allowed to cool. The dry matter of leaf samples was calculated. The dry matter samples were placed in a muffle furnace at 600 °C for two hours (at this temperature all the organic matter burns, leaving white ash) and percent ash was calculated. Estimation of total nitrogen, phosphorus, and potassium was carried out for the leaves using the methods described in [21]. The total nitrogen of leaves was estimated by the Kjeldahl method. A 5 mL digested solution was steam distilled in the presence of 10 mL alkali (NaOH), and the ammonia released was passed into a 2 mL boric acid mixture, which was then titrated against standard N/100 HCl to a green end. Total nitrogen was calculated by the standard value, i.e., 1 mL of N/100 HCl = 0.14 mg of nitrogen. To record the crude protein in the foliage, total nitrogen was multiplied by a factor of 6.25. The phosphorus contents were estimated by using the digested solution and ammonium molybdate solution, which were dissolved in slightly warm distilled water and then filtered. Filtered extract was mixed with concentrated $H_2SO_4$ and this solution was mixed thoroughly with freshly prepared stannous chloride solution. After 30 min, the absorbance was measured at 700 nm using a DU-640B Beckman spectrophotometer. The reagent blank consisted of everything except the digested solution. Potassium chloride standard stock solution (KCl), analytical regent (AR) grade (dried at 105 °C for one hour), was dissolved in distilled water. Diluting the stock solution in distilled water was used for calibration of the instrument; working potassium standard solutions of 40 and 100 ppm were prepared. The digested solution was increased to the final volume by using distilled water in a 50 mL volumetric flask. Potassium content was determined using a CL-31 flame photometer unit from which the potassium in ppm was read directly.

For the estimation of calcium, dry calcium carbonate ($CaCO_3$) was dissolved in a minimum quantity of 1:1 hydrochloric acid (AR) and made up to 1 L of distilled water, which was used to prepare 40 and 100 ppm calcium standard solutions. Taking the digested solution with distilled water in a 50 mL volumetric flask, the final volume calcium filter was analyzed using the CL flame photometer unit, from which the calcium in ppm was read directly.

Similarly, soluble sugars and starch in leaf samples were determined [22]. Plant material (fine powder) was homogenized with 80% boiling alcohol. The homogenate was centrifuged at 2000 rpm for 10 min. The supernatants were used for the estimation of soluble sugars. The supernatant was placed in a dry test tube and cold anthrone reagent (0.2% anthrone in conc. $H_2SO_4$) was slowly added through the wall of the test tube and the contents were mixed with a cyclomixer. The mixed contents were kept in a boiling water bath for 7–8 min and then cooled to room temperature. Absorbance of the solutions was measured at 620 nm against a suitable blank of distilled water in a Du-640 Beckman spectrophotometer. The residue remaining was suspended in 52% (*v/v*) perchloric acid, centrifuged, and percent soluble starch in the supernatant was estimated by following the abovementioned anthrone reagent method by using a Du-640 Beckman spectrophotometer.

The crude fiber was estimated using the method described in [23]. Fine power was placed in a conical flask and 2.5% $H_2SO_4$ and distilled water were added into it. The mixture was boiled for 30 min and then filtered through Whatman No.1 filter paper over a funnel. The residue was washed in hot distilled water and transferred back into the beaker. NaOH (2.5%) was added to it. Finally, it was made to volume by adding distilled water and gently heated for 30 min. This mixture was again filtered and washed with hot distilled water. The residue was finally washed with a small amount of 95% NaOH. The residue was transferred to a silica crucible and dried in an electric oven at 100 °C constant temperature for two hours. The crucible was cooled in the desiccator and the contents were weighed. The content was ignited in the muffle furnace until the formation of white ash. It was then cooled and weighed. The white content is called crude fiber and it was calculated.

Total phenolics were assessed as per the procedure of Makker et al. [24]. Plant material was crushed with a pestle and mortar in 95% warm ethanol. The homogenate was centrifuged at 10,000 rpm for 20 min. The supernatant was made up to volume with ethanol. The extract was taken in a test tube and the final volume was made with distilled water.

Folin–Ciocalteu reagent was added and the mixture was shaken properly. After 3 min, sodium carbonate reagent was added to the above mixture and again shaken thoroughly. The contents of the tubes were thoroughly mixed using a vortex mixer and kept at room temperature for 40 min. Absorbance of the contents was recorded at 725 nm using a DU 640 Beckman spectrophotometer. The amount of total phenolics was calculated as standard tannic acid (a fresh prepared solution was always used) equivalent to the calibration curve and the phenolic content (%) was expressed on a dry matter basis.

### 2.4. Statistical Analysis

Windows-based statistics software (Microsoft Excel) was used for calculating the standard deviation and correlation coefficient. Analysis of variation (ANOVA) randomized block design (RBD) and a two-factor factorial experiment was estimated by using the statistical software package WASP—Web Agri Stat Package IGAR Goa. The critical difference was compared for evaluating significant ($p < 0.05$) variations in nutritive value between different altitude and season. For the correlation coefficient, significant levels were observed at $p < 0.01$ and $p < 0.05$.

## 3. Results

The leaves collected from the Srinagar population exhibited significantly higher dry matter than those from Kandikhal. There was a significant inverse correlation between dry matter content and altitude of foliage, which indicated a decreasing trend in dry matter with increasing altitude (Table 2). Significant ($p < 0.05$) monthly variations were recorded for the dry matter content, irrespective of the populations. Dry matter content was the highest (53.28%) in December and the lowest (21.08%) in March, resulting in a mean value of 39.24% (Table A1).

The ash content of Agroda's tree population was significantly ($p < 0.05$) greater than that from Srinagar. Concerning altitudinal variation, ash content was the maximum (16.90%) for Agroda and the minimum (15.09%) for the Srinagar tree population. It was nonsignificant and positively correlated with altitude (Table 2). Significant monthly variations were recorded for ash content in all the populations. On average, the ash content in the foliage was the highest in October and the lowest in March, with mean values of 18.70 and 12.80%, respectively (Table A1). Nitrogen content was significant ($p < 0.01$) and positively correlated with altitude. The maximum nitrogen (2.24%) was observed in the population of Badiyargaon and the minimum (2.04%) in Srinagar (Table 2). Nitrogen variation was also significant ($p < 0.05$) during the months. The maximum nitrogen content (2.71%) was recorded in May and the minimum (1.46%) in March (Table A1).

Phosphorus content in the foliage of the Srinagar population was significantly greater compared to that of Kandikhal. There was a significant ($p < 0.05$) inverse correlation between phosphorus content and altitude of foliage, indicating a decreasing trend for phosphorus with increasing altitude (Table 2). Significant monthly variations were recorded for phosphorus, with the highest mean value in May (0.16%) and the lowest in September (0.08%). Comparing the phosphorus content between seasons and among populations, the maximum was reported in May for all the sources (Table A2). There was a significant positive correlation between potassium content and altitude of foliage, indicating an increasing trend for potassium content in the foliage with increasing altitude. On average, the Badiyargaon population exhibited the highest (0.50%) and Khandikhal showed the lowest (0.42%) potassium content (Table 2). All the populations exhibited the maximum potassium content in March, with mean value of 0.76% (Table A2). On average, the high altitudinal Badiyargaon tree population showed the highest (3.26%) and Srinagar the lowest (2.87%) calcium content; however, the average calcium content in the foliage was recorded as 3.10%, irrespective of altitudinal and monthly variations. There was significant positive correlation between calcium content and altitude of foliage collection (Table 2). All the populations exhibited higher values of calcium in December (4.31%) and lower values in July (2.33%). Calcium content varied significantly among different populations (Table A2).

On average, the Srinagar tree population at low altitude showed the highest soluble sugar content (1.67%), irrespective of monthly variations, whereas, at high (1980 masl) altitude, the Badiyargaon population exhibited the lowest (1.44%). Regardless of the population and monthly variation, the average value of the soluble sugars was recorded as 1.56%. There was a significant inverse correlation between soluble sugar content and the elevational range of foliage, indicating a decreasing trend for sugar with increasing altitude (Table 2). Data on soluble sugars also exhibited significant monthly and altitudinal variations. However, concerning monthly variations, there was no definite trend in soluble sugars; viz., the foliage of Srinagar, Kandikhal, and Badiyargaon had peak values in October, but the foliage of Agroda exhibited a peak value in May (Table A3). Among various populations, Kandikhal had the highest (6.31%) and Agroda the lowest (4.71%) starch content, irrespective of monthly variations. There was a positive significant correlation between starch content and altitude of foliage, which indicates that starch in the foliage increased with increase in altitude (Table 2). Regardless of the population, the maximum starch content of collected foliage was recorded in December (6.53%), and the minimum was recorded in July (4.79%). There was significant variation in all the seasons (Table A3).

For crude protein, the Badiyargaon population had the highest value (14.02%) and Srinagar the lowest value (12.66%). On average, the crude protein content in *C. australis* was recorded as 13.18%, irrespective of monthly and altitudinal variations. There was significant positive correlation between crude protein content and altitude of the foliage (Table 2). Significant monthly variations were observed for crude protein in the foliage, irrespective of population. On average, crude protein was the highest for all the populations in May with a mean value of 16.97% and the lowest in March with a mean value of 9.17% (Table A4).

For crude fiber, the Khandikhal population had the highest (17.11%) and Badiyargaon the lowest (16.08%). On average, the crude fiber in *C. australis* was recorded as 16.51%, irrespective of monthly and altitudinal variations. There was no significant effect of altitude on crude fiber content that was recorded (Table 2). Data on crude fiber also exhibited significant monthly variations. All the populations had high crude fiber content in July with a mean value of 19.80% (Table A4). Among various populations, Khandikhal had the maximum phenolic content (0.78%) and Srinagar the minimum (0.42%), irrespective of monthly variations. Phenolic content did not exhibit any significant relationship with altitude (Table 2). There was significant monthly variation for the phenolic content. Irrespective of altitudinal variations, phenolic content was maximum (0.72%) in November and minimum (0.52%) in March and April (Table A4).

Significant ($p < 0.01$ and $p < 0.05$), positive correlation was recorded between altitude vs. nitrogen, calcium, crude protein, potassium, and starch, and inversely correlated with dry matter, phosphorus, and sugar. Latitude showed significant ($p < 0.01$) positive correlation with nitrogen, crude protein, and starch, and significant ($p < 0.05$) positive correlation with potassium and calcium, while there was significant ($p < 0.01$) inverse correlation with dry matter, total ash, phosphorus, and sugar. Dry matter, phosphorus, and potassium showed significant positive ($p < 0.05$) correlation with longitude, while longitude showed a significant ($p < 0.01$ and $p < 0.05$) negative relationship with crude fiber, starch, and phenolic content. Temperature had a significant ($p < 0.01$) positive correlation with dry matter, phosphorus, and sugar. A significant negative ($p < 0.01$ and $p < 0.05$) correlation was also recorded between temperature vs. nitrogen, calcium, crude protein, potassium, and starch. Rainfall was significantly ($p < 0.01$ and $p < 0.05$) positively correlated with nitrogen, potassium, calcium, and crude protein, while inversely significantly ($p < 0.01$ and $p < 0.05$) correlated with dry matter, total ash, phosphorus, and sugar. Soil pH was significantly ($p < 0.05$) positively correlated with potassium and calcium, and negatively significantly correlated with crude fiber (Table 2).

The nutrient contents in *C. australis* foliage collected from different populations and seasons showed significant variation ($p < 0.01$) during the different seasons, except for total ash content (Table 3).

**Table 2.** Nutritive values of *C. australis* foliage as influenced by altitude, irrespective of seasonal variation (±SD); Means followed by the same letter are not significantly ($p < 0.05$) different.

| Altitude (masl) | Dry Matter (%) | Total Ash (%) | Nitrogen (%) | Phosphorus (%) | Potassium (%) | Calcium (%) | Crude Protein (%) | Sugar (%) | Starch (%) | Crude Fiber (%) | Phenolic (%) |
|---|---|---|---|---|---|---|---|---|---|---|---|
| 550 | 41.48 a ± 11.36 | 16.36 ab ± 1.86 | 2.04 c ± 0.42 | 0.11 a ± 0.03 | 0.44 ab ± 0.15 | 2.87 a ± 0.55 | 12.66 a ± 2.58 | 1.66 a ± 0.28 | 4.93 ab ± 0.62 | 1.64 a ± 0.29 | 0.43 a ± 0.10 |
| 1180 | 41.25 ab ± 8.33 | 16.90 b ± 1.92 | 2.05 c ± 0.37 | 0.11 a ± 0.02 | 0.43 b ± 0.19 | 3.08 ab ± 0.66 | 12.69 a ± 2.38 | 1.63 ab ± 0.18 | 4.71 a ± 0.75 | 1.64 ab ± 0.19 | 0.78 b ± 0.15 |
| 1550 | 36.81 c ± 10.22 | 16.58 ab ± 2.39 | 2.14 b ± 0.40 | 0.10 b ± 0.03 | 0.42 b ± 0.20 | 3.16 b ± 0.59 | 13.34 ab ± 2.48 | 1.49 bc ± 0.18 | 6.31 c ± 0.77 | 1.71 b ± 0.16 | 0.78 b ± 0.15 |
| 1980 | 37.06 bc ± 10.23 | 16.08 a ± 1.76 | 2.24 a ± 0.41 | 0.10 b ± 0.02 | 0.50 a ± 0.18 | 3.26 b ± 0.64 | 14.02 b ± 2.58 | 1.45 c ± 0.26 | 6.12 bc ± 0.82 | 1.61 a ± 0.24 | 0.52 ab ± 0.16 |
| "r" | −0.64 * | 0.11 ns | 0.92 ** | −0.67 * | 0.57 * | 0.90 ** | 0.91 ** | −0.94 ** | 0.80 ** | −0.11 ns | 0.37 ns |
| Latitude | −0.88 ** | −0.82 ** | 0.91 ** | −0.91 ** | 0.63 * | 0.62 * | 0.92 ** | −0.89 ** | 0.91 ** | 0.01 ns | −0.29 ns |
| Longitude | 0.57 * | −0.28 ns | −0.12 ns | 0.53 * | 0.56 * | −0.27 ns | −0.12 ns | 0.40 ns | −0.60 * | −0.95 ** | −0.64 * |
| Temperature | 0.84 ** | 0.30 ns | −0.90 ** | 0.83 ** | −0.56 * | −1.00 ** | −0.89 ** | 0.92 ** | −0.74 * | 0.07 ns | −0.31 ns |
| Rainfall | −0.78 * | −0.60 * | 0.97 ** | −0.79 * | 0.81 ** | 0.90 ** | 0.97 ** | −0.89 ** | 0.71 * | −0.33 ns | −0.05 ns |
| pH | −0.18 ns | 0.10 ns | 0.44 ns | −0.17 ns | 0.49 * | 0.70 * | 0.41 ns | −0.35 ns | 0.03 ns | −0.50 * | 0.33 ns |

* Significant at $p < 0.05$, ** Significant at $p < 0.01$, ns = nonsignificant.

**Table 3.** Two-way analysis of variance for altitude and month of foliage collection.

| Source of Variation | Degrees of Freedom | Dry Matter (%) | Total Ash (%) | Nitrogen (%) | Phosphorus (%) | Potassium (%) | Calcium (%) | Crude Protein (%) | Sugar (%) | Starch (%) | Crude Fiber (%) | Phenolic (%) |
|---|---|---|---|---|---|---|---|---|---|---|---|---|
| | | | | | | F-Value | | | | | | |
| Altitude | 3 | 67.11 ** | 1.06 ** | 3.79.05 ** | 7.72 ** | 78.13 ** | 136.99 ** | 8556.94 ** | 263.95 ** | 4045.70 ** | 100.23 ** | 1024.08 ** |
| Month | 9 | 443.48 ** | 23.78 ns | 2421.57 ** | 35.33 ** | 661.92 ** | 762.58 ** | 50,294.32 ** | 470.06 ** | 824.47 ** | 573.97 ** | 79.73 ** |
| Altitude × Month | 27 | 3.30 ** | 2.72 ** | 37.48 ** | 3.62 ** | 54.72 ** | 6.96 ** | 720.87 ** | 74.79 ** | 460.51 ** | 92.28 ** | 71.50 ** |

** Significant at $p < 0.01$, ns = nonsignificant.

## 4. Discussion

The present study revealed that altitude and season significantly influenced the nutritive values of *C. australis* foliage. Crude protein, phosphorus, and crude fiber exhibited peak values during the summer season (May–June); the other parameters showed optimum values during winter (i.e., October–December). Many workers have also reported seasonal variations in the concentration of nutrients in tree foliage [25–29]. Altitudinal variation in leaf chemistry and nutritive values of other fodder species is also well documented [30–33] in different parts of another region. However, little work has been conducted on the effects of altitude and seasons in *C. australis* foliage, particularly in the Garhwal Himalaya region. Morecroft et al. [34] reported that nitrogen concentration in plants increased with altitudinal gradient. In *C. australis* foliage, a significant ($p < 0.01$) positive correlation between altitude of foliage collection and crude protein was recorded, which seems in agreement with earlier findings by Morecroft [34]. In general, crude protein content has been reported to be the most important nutrient, which is generally taken as the index of nutritive value [17]. During the course of the present investigation, significant monthly variations in crude protein content were recorded. Variation in protein content may partly be attributed to retranslocation of leaf nitrogen into branches before leaf fall and partly due to a dilution factor with expansion and maturity of the leaves, and the former strategy is adopted by the plants for conserving nitrogen [35,36].

In *C. australis,* more than 50% of annual leaf biomass production was completed during March to May and the remaining during June to September. Further very rapid leaf expansion after bud burst has been reported in this species, and leaves attain full size within 10–15 days after bud burst (personal observations of the authors). It might be possible because of high protein content in *C. australis* foliage during May. The leaf longevity of *C. australis* foliage was recorded about 180 days. In an earlier study, Khosla et al. [35] recorded high crude protein in young leaves of *C. australis*, which decreased remarkably with leaf maturation. In the present investigation, the highest crude protein (17.00%) was recorded in May and the lowest (9.2%) in March, irrespective of provenance variation. Another study by Verma et al. [37] also showed significant decrease in nitrogen concentration of *C. australis* foliage with leaf maturation. Kumar and Toky [38] also observed significant variations in nitrogen, phosphorus, potassium, calcium, and magnesium in the foliage of *Albizia lebbek* provenances. Seasonal variations for ash, crude protein, and protein precipitation of *Artocarpus lakoocha* and *Quercus semecarpifolia* foliage have also been recorded by Wood et al. [39]. Various workers have also recorded that total phenolic and chlorophyll contents and carotenoids in plants decreased with increasing altitude [40,41]. Dry matter, ash content, crude fiber, phenolic content, calcium, starch, and soluble sugars increased with leaf maturation in *C australis*. These findings are in agreement with those of Subba et al. [15,42]. Ash content, dry matter, and crude fiber increased in *C. australis* foliage with leaf maturation. Wood et al. [43] have also found similar observations in 13 tree fodder species of Nepal, Himalaya. Thus, the nutritive value of *C. australis* foliage is strongly influenced by provenance (altitude). The dry matter and ash contents were significantly varied among the altitudes. Dhakal et al. [44] also reported that dry mater and ash contents of *Melia azadarch*, *Morus alba*, *Ficus roxburghii,* and *Ficus nemoralis* were significantly varied among the altitudes. Potassium, calcium, crude protein, and starch increased significantly in *C. australis* foliage with increasing altitude, whereas, dry matter, phosphorus, and soluble sugars decreased with increasing altitude in *C. australis* foliage (Table 2). The crude protein was significantly ($p < 0.05$) varied among the altitudes. Similar findings were recorded by Koutsoukis et al. [45], and they reported that crude protein was significantly ($p < 0.05$) varied among the altitudinal zones for all botanical plant groups in subalpine grassland. The decreasing concentration of various parameters with increasing altitude may be attributed to the fact that, generally, nitrogen is taken up by plants from the soil in spring or, at the same time, remobilized from reserves in over-wintering tissues (rhizomes and roots), whereas, biomass is produced throughout the growing season and progressively dilutes the nitrogen [46]. As compared to nitrogen concentration, very few studies are available on the concentration of phosphorus, potassium, calcium, and magnesium along altitudinal gradients. However, Korner [47] reported that high altitude species of *Alps* had higher concentration of phosphorus and magnesium but did not find any

difference in the concentrations of potassium and calcium. However, in the present study, potassium and calcium was significantly positively correlated with altitude. On the other hand, Harrison et al. [48] found decreasing phosphorus concentrations with increasing altitude in British grassland swards, which supports the present findings.

In the present study, foliage collected from higher altitudes showing higher nutrient contents might be due to rich soil nutrient status in this region as compared to the other sites. The higher soil nutrients at higher altitudes is because of the higher decomposition rate of humus due to the presence of higher moisture contents in the soils and the moderate temperature at the site [49,50]. The soil nutrients vary due to variations in vegetation, altitude, and land use and soil heterogeneity [51–53]. Nitrogen, potassium, calcium, crude protein, and starch are positively significantly correlated with latitude, indicating that their contents increase toward the northern extremes. Dry matter, ash contents, phosphorus, and sugar correlate inversely with latitude, indicating that these contents increase toward the southern extremes. Longitude exhibited a positive correlation with the dry matter, phosphorus, and potassium, indicating that this trait increases toward the eastern extremes. The inverse correlation of longitude with starch, crude fiber, and phenolics indicates that these traits increase toward western extremes. Temperature significantly increased the dry matter, phosphorus, and sugar while decreasing the nitrogen, potassium, calcium, crude protein, and starch. Rainfall significantly decreased the dry matter, total ash, phosphorus, and sugar while increasing the nitrogen, potassium, calcium crude protein, and starch. Soil pH of sites increases the potassium and calcium and decreases the crude fiber. So far, no attempts have been made to evaluate the fodder quality of this potential tree crop as influenced by altitudinal and seasonal variations. However, morphological, agronomic, and biochemical parameters have been used widely in the evaluation of various tree crops. Based on these findings, suitable populations could be recommended for a mass afforestation program in various agroforestry systems for quality fodder production.

## 5. Conclusions

The nutritive value of *C. australis* leaves is strongly influenced by different seasons and altitudes. The high altitudinal population exhibited higher potassium, calcium, crude protein, and starch, and lower levels of phenolic, as compared to lower altitude. Ash content, dry matter and crude fiber increased in *Celtis* foliage with leaf maturation. Irrespective of altitudinal variation, dry matter, ash content, calcium, starch, and sugar content exhibited the highest values during November–December (winter season) and potassium, phosphorus, crude protein, and crude fiber were the greatest during May–June (summer season). The minimum value of phenolics was recorded during July–September (rainy season) in *Celtis* foliage. Thus, *C. australis* mass propagation should be carried out for reforestation in degraded lands, wastelands, and agricultural lands at middle to higher altitudes. This provide nutritive fodder and mitigate the problem of insufficient palatable nutritive tree fodder during peak periods, thus improving the diet and health of the cattle population in Himalayan hilly region of India.

**Author Contributions:** Conceptualization, B.S.; Formal analysis, B.S.; Supervision, B.P.B.; Writing–original draft, B.S.; Writing–review & editing, M.K., M.M.S.C.-P. and B.P.B. All authors have read and agreed to the published version of the manuscript.

**Funding:** No external funding received for this research.

**Informed Consent Statement:** The authors confirm that they have not used animals in this study.

**Data Availability Statement:** All data generated or analysed during this study are included in this summited article.

**Acknowledgments:** Authors are thankful to the Department of Forestry, and High Altitude Plant Physiology Research Centre, H.N.B. Garhwal University Srinagar Garhwal for providing laboratory facilities.

**Conflicts of Interest:** Authors declared that no conflict of interest.

## Appendix A

**Table A1.** Altitudinal and monthly variation in dry matter (DM), total ash (%), and nitrogen (%) in *C. australis* foliage (±SD). Means followed by the same letter are not significantly (*p* < 0.05) different.

| Month | Altitude (masl) | | | | | | | | | | | |
|---|---|---|---|---|---|---|---|---|---|---|---|---|
| | 550 | | | 1180 | | | 1550 | | | 1980 | | |
| | Dry Matter (%) | Total Ash (%) | Nitrogen (%) | Dry matter (%) | Total Ash (%) | Nitrogen (%) | Dry Matter (%) | Total Ash (%) | Nitrogen (%) | Dry Matter (%) | Total Ash (%) | Nitrogen (%) |
| March | 20.00 ± 2.50 [g] | 13.01 ± 1.22 [i] | 1.42 ± 0.02 [h] | 26.98 ± 1.95 [f] | 12.42 ± 2.13 [j] | 1.32 ± 0.02 [i] | 18.72 ± 2.5 [g] | 11.46 ± 1.65 [i] | 1.69 ± 0.09 [i] | 18.77 ± 2.02 [f] | 12.87 ± 1.95 [h] | 1.46 ± 0.02 [i] |
| April | 27.30 ± 1.98 [f] | 14.82 ± 2.14 [g] | 2.04 ± 0.03 [b] | 31.42 ± 1.95 [e] | 15.48 ± 1.69 [i] | 2.30 ± 0.02 [b] | 26.16 ± 2.32 [f] | 14.41 ± 1.98 [h] | 2.35 ± 0.03 [d] | 26.14 ± 1.96 [g] | 14.82 ± 1.85 [g] | 2.26 ± 0.05 [f] |
| May | 38.50 ± 2.52 [e] | 16.82 ± 1.04 [e] | 2.69 ± 0.02 [a] | 36.26 ± 1.32 [d] | 16.65 ± 1.68 [h] | 2.57 ± 0.02 [a] | 32.32 ± 2.51 [e] | 15.42 ± 1.99 [g] | 2.77 ± 0.02 [a] | 32.75 ± 1.99 [f] | 15.08 ± 1.76 [f] | 2.82 ± 0.03 [a] |
| June | 40.40 ± 2.12 [de] | 17.21 ± 2.15 [d] | 2.45 ± 0.02 [b] | 38.92 ± 1.25 [cd] | 16.80 ± 1.95 [g] | 2.22 ± 0.03 [c] | 34.56 ± 1.65 [e] | 16.22 ± 1.92 [e] | 2.58 ± 0.03 [b] | 33.35 ± 1.52 [f] | 15.84 ± 1.98 [e] | 2.68 ± 0.03 [b] |
| July | 40.00 ± 2.51 [de] | 17.24 ± 2.12 [d] | 2.38 ± 0.03 [c] | 39.12 ± 1.62 [c] | 17.07 ± 1.75 [f] | 2.32 ± 0.02 [b] | 34.45 ± 2.52 [e] | 17.64 ± 1.89 [d] | 2.43 ± 0.03 [c] | 36.87 ± 1.68 [e] | 16.66 ± 2.05 [d] | 2.54 ± 0.02 [c] |
| August | 41.74 ± 2.02 [d] | 17.64 ± 2.05 [c] | 2.17 ± 0.03 [d] | 44.17 ± 1.95 [b] | 17.25 ± 1.85 [e] | 2.16 ± 0.07 [d] | 37.88 ± 2.45 [d] | 18.07 ± 1.79 [c] | 2.20 ± 0.04 [e] | 39.54 ± 1.95 [d] | 17.82 ± 2.14 [b] | 2.40 ± 0.02 [d] |
| September | 46.75 ± 2.05 [c] | 18.23 ± 2.10 [b] | 2.07 ± 0.03 [e] | 45.50 ± 1.85 [b] | 17.84 ± 1.98 [d] | 2.11 ± 0.03 [e] | 39.87 ± 2.34 [d] | 18.87 ± 1.83 [b] | 2.03 ± 0.04 [f] | 42.12 ± 1.69 [d] | 18.24 ± 2.15 [a] | 2.30 ± 0.02 [e] |
| October | 50.00 ± 2.62 [b] | 18.86 ± 1.95 [a] | 1.90 ± 0.02 [f] | 46.70 ± 1.65 [b] | 18.43 ± 2.24 [b] | 1.91 ± 0.02 [f] | 44.54 ± 2.81 [c] | 19.43 ± 1.53 [a] | 1.94 ± 0.03 [g] | 45.12 ± 1.98 [c] | 18.23 ± 2.04 [a] | 2.21 ± 0.02 [g] |
| November | 55.22 ± 2.52 [a] | 16.20 ± 1.68 [f] | 1.84 ± 0.03 [g] | 51.72 ± 1.32 [a] | 19.41 ± 2.01 [a] | 1.86 ± 0.02 [g] | 48.12 ± 2.51 [a] | 18.06 ± 1.53 [c] | 1.84 ± 0.04 [h] | 48.44 ± 1.85 [b] | 16.81 ± 1.98 [c] | 1.97 ± 0.03 [h] |
| December | 55.62 ± 2.54 [a] | 14.08 ± 1.95 [h] | 1.42 ± 0.02 [h] | 52.24 ± 1.52 [a] | 18.00 ± 2.51 | 1.63 ± 0.02 [h] | 53.31 ± 2.62 [a] | 16.02 ± 1.89 [f] | 1.55 ± 0.03 [j] | 52.35 ± 1.98 [a] | 14.84 ± 1.86 [g] | 1.79 ± 0.03 [i] |

**Table A2.** Provenance and monthly variation in potassium, phosphorus, and calcium (%) in *C. australis* foliage (±SD). Means followed by the same letter are not significantly (*p* < 0.05) different.

| Month | Altitude (masl) | | | | | | | | | | | |
|---|---|---|---|---|---|---|---|---|---|---|---|---|
| | 550 | | | 1180 | | | 1550 | | | 1980 | | |
| | Potassium (%) | Phosphorus (%) | Calcium (%) | Potassium (%) | Phosphorus (%) | Calcium (%) | Potassium (%) | Phosphorus (%) | Calcium (%) | Potassium (%) | Phosphorus (%) | Calcium (%) |
| March | 0.71 ± 0.06 [a] | 0.11 ± 0.02 [bc] | 2.37 ± 0.39 [g] | 0.77 ± 0.08 [a] | 0.12 ± 0.01 [b] | 2.55 ± 0.58 [g] | 0.78 ± 0.06 [a] | 0.13 ± 0.03 [b] | 2.53 ± 0.29 [f] | 0.78 ± 0.04 [a] | 0.11 ± 0.02 [cd] | 2.73 ± 0.28 [f] |
| April | 0.63 ± 0.06 [b] | 0.15 ± 0.04 [a] | 2.83 ± 0.65 [de] | 0.73 ± 0.08 [a] | 0.14 ± 0.01 [b] | 3.19 ± 0.64 [d] | 0.77 ± 0.06 [a] | 0.13 ± 0.01 [b] | 3.25 ± 0.47 [c] | 0.77 ± 0.08 [a] | 0.12 ± 0.01 [bc] | 3.38 ± 0.41 [c] |
| May | 0.38 ± 0.03 [e] | 0.17 ± 0.03 [a] | 3.09 ± 0.62 [c] | 0.34 ± 0.04 [d] | 0.17 ± 0.03 [a] | 3.42 ± 0.51 [c] | 0.30 ± 0.08 [e] | 0.16 ± 0.02 [a] | 3.61 ± 0.49 [b] | 0.42 ± 0.05 [e] | 0.15 ± 0.02 [a] | 3.46 ± 0.46 [c] |
| June | 0.34 ± 0.05 [f] | 0.09 ± 0.01 [d] | 2.60 ± 0.44 [f] | 0.32 ± 0.08 [d] | 0.10 ± 0.01 [bc] | 2.45 ± 0.25 [g] | 0.22 ± 0.06 [f] | 0.08 ± 0.01 [d] | 2.88 ± 0.27 [d] | 0.36 ± 0.05 [f] | 0.09 ± 0.03 [de] | 2.88 ± 0.58 [e] |
| July | 0.26 ± 0.04 [g] | 0.11 ± 0.02 [bc] | 2.29 ± 0.42 [g] | 0.25 ± 0.04 [e] | 0.09 ± 0.03 [bc] | 2.13 ± 0.27 [g] | 0.34 ± 0.04 [d] | 0.11 ± 0.01 [c] | 2.53 ± 0.38 [f] | 0.51 ± 0.06 [c] | 0.11 ± 0.01 [cd] | 2.37 ± 0.38 [g] |
| August | 0.25 ± 0.04 [g] | 0.12 ± 0.03 [b] | 2.36 ± 0.30 [g] | 0.23 ± 0.04 [e] | 0.08 ± 0.01 [c] | 2.66 ± 0.59 [f] | 0.29 ± 0.05 [e] | 0.07 ± 0.01 [de] | 2.77 ± 0.39 [e] | 0.37 ± 0.05 [g] | 0.08 ± 0.02 [e] | 2.78 ± 0.35 [f] |
| September | 0.47 ± 0.04 [cd] | 0.09 ± 0.02 [d] | 2.73 ± 0.52 [ef] | 0.23 ± 0.04 [e] | 0.08 ± 0.02 [c] | 3.06 ± 0.57 [e] | 0.23 ± 0.06 [f] | 0.06 ± 0.01 [e] | 2.92 ± 0.27 [d] | 0.22 ± 0.09 [g] | 0.07 ± 0.01 [e] | 3.15 ± 0.58 [d] |
| October | 0.50 ± 0.02 [c] | 0.11 ± 0.02 [bc] | 2.98 ± 0.34 [cd] | 0.48 ± 0.09 [c] | 0.09 ± 0.01 [bc] | 3.35 ± 0.34 [c] | 0.43 ± 0.05 [b] | 0.10 ± 0.02 [c] | 3.11 ± 0.68 [c] | 0.53 ± 0.07 [c] | 0.09 ± 0.01 [de] | 3.36 ± 0.32 [c] |
| November | 0.40 ± 0.05 [e] | 0.07 ± 0.01 [e] | 3.44 ± 0.43 [b] | 0.47 ± 0.03 [c] | 0.09 ± 0.02 [bc] | 3.81 ± 0.54 [b] | 0.48 ± 0.06 [b] | 0.10 ± 0.01 [c] | 3.71 ± 0.50 [b] | 0.61 ± 0.09 [a] | 0.14 ± 0.01 [ab] | 3.96 ± 0.73 [b] |
| December | 0.46 ± 0.03 [d] | 0.09 ± 0.01 [d] | 4.05 ± 0.74 [a] | 0.52 ± 0.04 [b] | 0.10 ± 0.03 [bc] | 4.26 ± 0.72 [a] | 0.40 ± 0.06 [c] | 0.11 ± 0.03 [c] | 4.37 ± 0.44 [a] | 0.46 ± 0.06 [d] | 0.12 ± 0.01 [bc] | 4.57 ± 0.34 [a] |

**Table A3.** Provenance and monthly variation in sugar and starch (%) in *C. australis* foliage (±SD). Means followed by the same letter are not significantly ($p < 0.05$) different.

| Month | Altitude (masl) | | | | | | | |
|---|---|---|---|---|---|---|---|---|
| | 550 | | 1180 | | 1550 | | 1980 | |
| | Sugar (%) | Starch (%) | Sugar (%) | Starch (%) | Sugar (%) | Starch (%) | Sugar (%) | Starch (%) |
| March | 1.52 ± 0.11 e | 4.16 ± 0.31 g | 1.41 ± 0.18 e | 4.15 ± 0.14 e | 1.29 ± 0.16 f | 6.03 ± 0.59 e | 1.12 ± 0.12 g | 5.31 ± 0.77 f |
| April | 1.54 ± 0.12 e | 4.68 ± 0.29 ef | 1.59 ± 0.16 d | 6.61 ± 0.54 a | 1.37 ± 0.14 e | 6.14 ± 0.37 e | 1.84 ± 0.14 b | 6.26 ± 0.31 d |
| May | 1.98 ± 0.21 a | 5.84 ± 0.46 a | 1.96 ± 0.13 a | 5.63 ± 0.25 b | 1.64 ± 0.14 b | 5.83 ± 0.17 f | 1.54 ± 0.14 d | 5.10 ± 0.39 g |
| June | 1.85 ± 0.17 c | 5.56 ± 0.34 b | 1.45 ± 0.09 e | 5.66 ± 0.45 b | 1.49 ± 0.12 d | 5.49 ± 0.42 g | 1.27 ± 0.08 f | 5.61 ± 0.57 e |
| July | 1.68 ± 0.12 d | 4.91 ± 0.47 d | 1.62 ± 0.17 d | 4.71 ± 0.21 d | 1.52 ± 0.12 d | 5.11 ± 0.35 h | 1.37 ± 0.10 e | 4.42 ± 0.28 h |
| August | 1.42 ± 0.11 f | 4.72 ± 0.37 e | 1.70 ± 0.23 c | 4.10 ± 0.36 e | 1.46 ± 1.10 d | 6.34 ± 0.46 d | 1.01 ± 0.08 h | 6.81 ± 0.44 c |
| September | 0.97 ± 0.10 g | 3.96 ± 0.38 h | 1.44 ± 0.26 e | 3.45 ± 0.46 f | 1.21 ± 1.14 g | 6.89 ± 0.45 c | 0.86 ± 0.09 i | 5.30 ± 0.49 f |
| October | 1.95 ± 0.15 ab | 4.60 ± 0.25 f | 1.66 ± 0.13 d | 2.71 ± 0.15 g | 1.84 ± 0.18 a | 6.32 ± 0.56 d | 1.92 ± 0.13 a | 7.30 ± 0.34 c |
| November | 1.89 ± 0.12 bc | 5.35 ± 0.24 c | 1.73 ± 0.15 c | 4.67 ± 0.46 d | 1.56 ± 0.16 c | 7.36 ± 0.32 b | 1.89 ± 0.12 ab | 7.45 ± 0.32 b |
| December | 1.88 ± 0.21 bc | 5.51 ± 0.26 b | 1.82 ± 0.12 b | 5.39 ± 0.37 c | 1.49 ± 0.14 d | 7.59 ± 0.22 a | 1.63 ± 0.15 c | 7.62 ± 0.41 a |

**Table A4.** Provenance and monthly variation in crude fiber, crude protein, and phenolic content (%) in *C. australis* foliage (±SD). Means followed by the same letter are not significantly ($p < 0.05$) different.

| Months | Altitude (masl) | | | | | | | | | | | |
|---|---|---|---|---|---|---|---|---|---|---|---|---|
| | 550 | | | 1180 | | | 1550 | | | 1980 | | |
| | Crude Fiber (%) | Crude Protein (%) | Phenolic (%) | Crude Fiber (%) | Crude Protein (%) | Phenolic (%) | Crude Fiber (%) | Crude Protein (%) | Phenolic (%) | Crude Fiber (%) | Crude Protein (%) | Phenolic (%) |
| March | 11.00 ± 3.56 e | 8.81 ± 1.56 i | 0.48 ± 0.04 de | 14.35 ± 2.32 J | 8.19 ± 1.21 j | 0.56 ± 0.07 ef | 15.25 ± 2.01 f | 10.56 ± 1.23 i | 0.65 ± 0.04 c | 15.15 ± 3.52 f | 9.13 ± 1.53 j | 0.65 ± 0.06 cde |
| April | 13.95 ± 2.96 d | 12.63 ± 1.86 f | 0.46 ± 0.05 e | 15.91 ± 3.21 f | 14.39 ± 1.52 c | 0.60 ± 0.08 de | 15.15 ± 2.09 g | 14.69 ± 1.52 d | 0.64 ± 0.07 cd | 15.75 ± 4.27 e | 14.13 ± 1.85 e | 0.62 ± 0.08 d |
| May | 16.21 ± 2.62 c | 16.75 ± 1.23 a | 0.41 ± 0.04 e | 18.05 ± 2.24 c | 16.19 ± 1.28 a | 0.36 ± 0.08 g | 16.45 ± 3.05 e | 17.31 ± 1.56 a | 0.31 ± 0.06 f | 16.95 ± 4.23 d | 17.63 ± 1.62 a | 0.38 ± 0.04 f |
| June | 16.56 ± 2.65 c | 15.25 ± 1.90 b | 0.49 ± 0.09 d | 16.94 ± 3.45 d | 14.13 ± 1.64 d | 0.51 ± 0.08 f | 18.78 ± 3.65 b | 16.16 ± 1.68 b | 0.56 ± 0.02 e | 20.05 ± 4.62 a | 16.75 ± 1.82 b | 0.55 ± 0.06 e |
| July | 21.55 ± 2.95 a | 14.69 ± 1.52 b | 0.71 ± 0.09 b | 18.82 ± 2.32 b | 14.69 ± 1.62 b | 0.73 ± 0.09 b | 19.74 ± 2.81 a | 15.12 ± 1.85 c | 0.89 ± 0.03 a | 19.15 ± 2.95 b | 15.88 ± 1.62 c | 0.84 ± 0.08 a |
| August | 19.45 ± 2.65 b | 13.50 ± 1.35 d | 0.78 ± 0.08 a | 16.35 ± 2.56 e | 13.22 ± 1.92 e | 0.76 ± 0.05 a | 18.65 ± 2.14 b | 13.81 ± 1.61 e | 0.75 ± 0.02 b | 18.15 ± 2.12 c | 15.34 ± 1.32 d | 0.72 ± 0.04 bc |
| September | 17.85 ± 4.62 b | 12.94 ± 1.86 c | 0.68 ± 0.08 b | 19.55 ± 2.62 a | 12.94 ± 1.35 f | 0.71 ± 0.05 c | 18.22 ± 3.51 c | 12.63 ± 1.62 e | 0.68 ± 0.09 c | 13.54 ± 3.95 g | 14.38 ± 1.25 e | 0.65 ± 0.06 cde |
| October | 16.82 ± 2.36 c | 11.75 ± 1.56 g | 0.52 ± 0.05 cd | 14.55 ± 2.53 i | 11.75 ± 1.64 g | 0.56 ± 0.04 ef | 16.34 ± 3.09 e | 12.06 ± 1.23 f | 0.59 ± 0.04 de | 13.47 ± 4.24 g | 13.81 ± 1.62 g | 0.62 ± 0.08 d |
| November | 16.34 ± 4.23 c | 11.44 ± 1.45 h | 0.54 ± 0.04 c | 15.15 ± 2.36 g | 11.5 ± 1.52 h | 0.66 ± 0.04 cd | 17.15 ± 2.98 d | 11.44 ± 1.52 h | 0.62 ± 0.06 d | 13.36 ± 4.20 g | 12.31 ± 1.42 h | 0.56 ± 0.04 e |
| December | 14.15 ± 2.32 d | 8.81 ± 1.23 i | 0.72 ± 0.06 b | 14.65 ± 2.35 h | 10.24 ± 1.29 i | 0.74 ± 0.06 a | 16.45 ± 3.54 e | 9.69 ± 1.53 j | 0.74 ± 0.07 bc | 15.35 ± 3.54 e | 11.19 ± 1.81 i | 0.76 ± 0.06 ab |

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
