# Peer review of "Seasonal and Altitudinal Variation in Chemical Composition of Celtis australis L. Tree Foliage"

_land, doi:10.3390/land11122271_

Round 1

Reviewer 1 Report

Dear Authors, I am honored to review your valued manuscript titled "Seasonal and altitudinal variation in chemical composition of Celtis australis L. tree foliage". 

I may kindly ask you to improve the introduction section by indicating the reasons why you needed to conduct that research, the problems that could not have been solved in the previous studies, the points make your study superior to the previous ones and add more references to enrich your literature screening. 

Please add also the map of the sites you sampled your samples. 

Please give more information about your study site such as climatic conditions, soil and parent rock properties, geomorphological structure, your sampling site land use, tree properties, tree population properties. 

One of the most important issue is the lack of the relation of your data to the soil nutrient element conditions which is an important determinant in terms of plant nutrition. 

Please see my uploaded file for further point base comments. 

Good luck. 

Author Response

Reviewer-1

Does the introduction provide sufficient background and include all relevant references?

Thank you for your positive comments

Are all the cited references relevant to the research?

Thank you for your positive comments

Is the research design appropriate?

Thank you for your positive comments

Are the methods adequately described?

Thank you for your positive comments

Are the results clearly presented?

Thank you for your positive comments

Are the conclusions supported by the results?

Yes

Improve the introduction section by indicating the reasons why you needed to conduct that research, the problems that could not have been solved in the previous studies, the points make your study superior to the previous ones and add more references to enrich your literature screening. 

Please add also the map of the sites you sampled your samples. 

Please give more information about your study site such as climatic conditions, soil and parent rock properties, geomorphological structure, your sampling site land use, tree properties, and tree population properties. 

One of the most important issues is the lack of the relation of your data to the soil nutrient element conditions which is an important determinant in terms of plant nutrition. 

Please see my uploaded file for further point base comments. 

Thank you for your positive word. Now all background information of the study have been given in the introduction section and also enrich literature of recent citations.

Map of the study area is now added of the study sites.

Sorry. but now all the informations have been added including climatic information, soil conditions, tree population properties of the sites etc.

Sorry, so far only nutritive values of the leaves in the present study are assessed. The data of soil analysis not taken for analysis therefore comparison cannot be made in the present study. However in future your suggestion will also be incorporated to improve the quality of manuscript for the comparison between nutritive value and soil.

Besides above comments, the comments on attached pdf of manuscript also improved for the quality of ms.

Reviewer 2 Report

The manuscript is suitable for publication in the journal with the corrections suggested in the text.

Author Response

Reviewer-2

Does the introduction provide sufficient background and include all relevant references?

Thank you for your positive comments

Are all the cited references relevant to the research?

Thank you for your positive comments

Is the research design appropriate?

Thank you for your positive comments

Are the methods adequately described?

Thank you for your positive comments

Are the results clearly presented?

Thank you for your positive comments

Are the conclusions supported by the results?

Yes

The manuscript is suitable for publication in the journal with the corrections suggested in the text.

Thank you for your positive comments on the ms and its suitability for publication in the Journal. Including that the highlighted information in the attached pdf also improved.

Reviewer 3 Report

Introduction

11. Four items of bibliography are far too few to convince the reader of the need and validity of the research. The authors practically limited in their description to information on the uses of Celtis australis.

22. The purpose of the research is unintelligible. The fact that the chemical composition of the leaves will vary at different times of the year is obvious. This is decided by different temperatures, rainfall, degree of leaf development. Similarly about the subject of height. Among other things, the chemical composition of leaves is affected by soil conditions, completely ignored by the authors.

Material and methods

1.     The information in this chapter is insufficient to reproduce the experiment.

2.     From how many trees were leaves collected? How many leaves were collected? How often were they collected - one at a time?

3.     The leaves should be dried in a dryer  (fixed temperature, fixed time)

4.     In how many replicates were the analyses performed?

5.     What does it mean "constant leaf moisture"?

6.     The description of the research methodology should not be limited to the literature source. What about the preparation of samples for analysis, for example.

7.     No description of soil and climatic conditions.

8.     The age of the trees? Method of cultivation?

9.     Years in which the experiment was conducted?

10.  Line 45: The full name of the species is required

Discussion

1.     Line 14. What species of trees?

2.     Lines 15-16. “Altitudinal variation in leaf chemistry and nutritive values of fodder species is also well documented 15 [14, 15, 16, 17]” If this is the case, why did the authors undertake this? (See the purpose of the study).

3.     The discussion is extensive. However, in my opinion this is because the chapter contains a lot of information that should be included in the Introduction chapter (e.g., lines 61-81). The authors should pay more attention to explaining why they believe nutrient content increases with increasing altitude. What is the impact of the growing season which is, after all, the object of the study.

Conclusion

1.     Line 91-94. The sentence is incomprehensible.

2.     Line 89.  The high altitudinal populations exhibited higher nutrients composition as compared to lower altitudeAll the nutrients compositions studied?

Abstract

It should be supplemented with, among others, information about the purpose of the study, the objects of the study. In its current form, it more constitute an extract from Results chapter.

References

Bibliography is poor and in its entirety older than 10 years. 

Author Response

Reviewer-3

Does the introduction provide sufficient background and include all relevant references?

Yes and now improved as per suggestions including relevant references

Are all the cited references relevant to the research?

Yes and improved

Is the research design appropriate?

Now the necessary inputs for reproduce of the study have been done

Are the methods adequately described?

Yes now improved

Are the results clearly presented?

Yes and improved

Are the conclusions supported by the results?

Yes and improved as required

   F  Introduction

        Four items of bibliography are far too few to convince the reader of the need and validity of the research. The authors practically limited in their description to information on the uses of Celtis australis.

Sorry. Now the necessary inputs have been given with relevant informations of concerned study.

The purpose of the research is unintelligible. The fact that the chemical composition of the leaves will vary at different times of the year is obvious. This is decided by different temperatures, rainfall, degree of leaf development. Similarly, about the subject of height. Among other things, the chemical composition of leaves is affected by soil conditions, completely ignored by the authors.

Sorry, so far only nutritive values of the leaves in the present study are assessed. The data of soil analysis not taken for analysis therefore comparison cannot be made in the present study. However in future your suggestion will also be incorporated to improve the quality of manuscript.

Material and methods

1.    The information in this chapter is insufficient to reproduce the experiment.

Sorry. Now the full details of the experiments have been added specially in methodology section for its clear understanding and reproduce of the study.

2.     From how many trees were leaves collected? How many leaves were collected? How often were they collected - one at a time?

Thank for your suggestion. The detailed information have been given in the methodology section.

3.     The leaves should be dried in a dryer  (fixed temperature, fixed time)

Yes both the options can be used either dryer or sunny days.

4.     In how many replicates were the analyses performed?

Information is given in methodology section

5.     What does it mean "constant leaf moisture"?

Sorry. It means complete dry of the leaves and no further moisture is reported

6.     The description of the research methodology should not be limited to the literature source. What about the preparation of samples for analysis, for example.

Details given in the methodology section

7.     No description of soil and climatic conditions.

Sorry.  Now the informations have been added.

8.     The age of the trees? Method of cultivation?

Usually phenotypically superior or mature age trees used for the sampling of experiment

9.     Years in which the experiment was conducted?

Usually the experiment was conducted long back but repeated to verify.

10.  Line 45: The full name of the species is required

Sorry, but now the informations have been corrected as required

Discussion

1. Line 14. What species of trees?

Pinus radiate, Spruces, Q Semicarpifolia and other experiments were conducted excluding C. australis etc.

2. Lines 15-16. “Altitudinal variation in leaf chemistry and nutritive values of fodder species is also well documented 15 [14, 15, 16, 17]” If this is the case, why did the authors undertake this?   (See the purpose of the study).

The documented work was carried out in  different parts of worlds and no such work carried out in the C. australis

3.   The discussion is extensive. However, in my opinion this is because the chapter contains a lot of information that should be included in the Introduction chapter (e.g., lines 61-81). The authors should pay more attention to explaining why they believe nutrient content increases with increasing altitude. What is the impact of the growing season which is, after all, the object of the study.

Thank you for your suggestions. The necessary modifications have been made as per your suggestions

Conclusion

1.   Line 91-94. The sentence is incomprehensible.

Sorry and needful modifications have been made

2.   Line 89.  “The high altitudinal populations exhibited higher nutrients composition as compared to lower altitude” All the nutrients compositions studied?

Sorry, but the data only studied as present in the results

Abstract

It should be supplemented with, among others, information about the purpose of the study, the objects of the study. In its current form, it more constitute an extract from Results chapter.

Thank you for your suggestion and necessary modification have been done/

References

Bibliography is poor and in its entirety older than 10 years

Sorry, now the recent citation have been added

Reviewer 4 Report

more information on relevant studies needs to be included in the introduction to build a strong background and highlight the research gap.  

Author Response

Reviewer4

Does the introduction provide sufficient background and include all relevant references?

Thank you for your positive comments

Are all the cited references relevant to the research?

Thank you for your positive comments

Is the research design appropriate?

Thank you for your positive comments

Are the methods adequately described?

Thank you for your positive comments

Are the results clearly presented?

Thank you for your positive comments

Are the conclusions supported by the results?

Thank you for your positive comments

The manuscript is suitable for publication in the journal with

The corrections suggested in the text

Thank you for your positive comments

More information on relevant studies needs to be included in the introduction to build a strong background and highlight the research gap.  

Thank you for suggestions and necessary inputs for the sufficient information of background in introduction section have been given now.

Round 2

Reviewer 1 Report

Dear Authors, I am thankful for your efforts to improve your manuscript at your best performance. May I kindly ask you to enter some soil nutrient elements data from the literature and previous studies to make more sound your results?

Additionally, the competition conditions are seriously efficient on the nutrient uptake of trees, thus can you please provide the site properties information to the Study Site section of the manuscript? The combination of the vegetative community is missing and this structure is highly determinant of nutrient uptake of target trees. 

You may check some of those manuscripts I gave below and others you may reach more easily that could be useful to refer: 

Khanday, M. U. D. D., Ram, D., Wani, J. A., & Ali, T. (2017). Vertical Distribution of Nutrient of the Soils of Namblan Sub-Catchment of Jhelum Basin of Srinagar District in Kashmir Valley. International Journal of Current Microbiology and Applied Sciences6(4), 375-381.

Sajed, S., & Munesh, K. (2014). Vegetation diversity and soil nutrient status of submergence zone of hydroelectric project in Srinagar of Garhwal Himalayas, India. International Journal of Biodiversity and Conservation6(12), 829-847.

Dervash, M. A., Lone, F. A., Bano, H., Wani, A. A., Khan, I., Wani, J. A., ... & Siddiqui, M. A. A. (2018). Assessment of physical properties of soil and organic carbon distribution in Srinagar city of Kashmir Himalaya. J. Pharmacog. and Phytochem, 7(6), 2145-2149. 

Sheikh, M. A., & Kumar, M. (2010). Nutrient status and economic analysis of soils in oak and pine forests in Garhwal Himalaya. Quercus1(1), 1600-1800.

Additionally, may I kindly ask you to publish the leaf total N content which may give elemental idea for the nutritive value of the leaves. 

Author Response

Comments

reply

Reviewer-1

Does the introduction provide sufficient background and include all relevant references?

Thank you for your positive comments

Are all the cited references relevant to the research?

Thank you for your positive comments

Is the research design appropriate?

Thank you for your positive comments

Are the methods adequately described?

Thank you for your positive comments

Are the results clearly presented?

Thank you for your positive comments

Are the conclusions supported by the results?

Thank you for your positive comments

·         Dear Authors, I am thankful for your efforts to improve your manuscript at your best performance.

·         Thank you for positive words.

·         May I kindly ask you to enter some soil nutrient elements data from the literature and previous studies to make more sound your results?

·         The soil related informations have been added for more sound the results.

·         Additionally, the competition conditions are seriously efficient on the nutrient uptake of trees, thus can you please provide the site properties information to the Study Site section of the manuscript? The combination of the vegetative community is missing and this structure is highly determinant of nutrient uptake of target trees. 

·         Sorry. But now necessary informations have been added in the manuscript

·         You may check some of those manuscripts I gave below and others you may reach more easily that could be useful to refer: 

Khanday, M. U. D. D., Ram, D., Wani, J. A., & Ali, T. (2017). Vertical Distribution of Nutrient of the Soils of Namblan Sub-Catchment of Jhelum Basin of Srinagar District in Kashmir Valley. International Journal of Current Microbiology and Applied Sciences6(4), 375-381.

Sajed, S., & Munesh, K. (2014). Vegetation diversity and soil nutrient status of submergence zone of hydroelectric project in Srinagar of Garhwal Himalayas, India. International Journal of Biodiversity and Conservation6(12), 829-847.

Dervash, M. A., Lone, F. A., Bano, H., Wani, A. A., Khan, I., Wani, J. A., ... & Siddiqui, M. A. A. (2018). Assessment of physical properties of soil and organic carbon distribution in Srinagar city of Kashmir Himalaya. J. Pharmacog. and Phytochem7(6), 2145-2149. 

Sheikh, M. A., & Kumar, M. (2010). Nutrient status and economic analysis of soils in oak and pine forests in Garhwal Himalaya. Quercus1(1), 1600-1800.

·         Additionally, may I kindly ask you to publish the leaf total N content which may give elemental idea for the nutritive value of the leaves. 

The following suggested necessary informations have been given in the manuscript.

Thank you for suggestions and necessary information have been now given.

Besides these informations,  the other queries have also been imporved as given in the pdf.

Reviewer 3 Report

I thank the authors for their positive response to the most of the comments and suggestions.

Author Response

Comments

reply

Reviewer-3

Does the introduction provide sufficient background and include all relevant references?

Thank you for your positive comments

Are all the cited references relevant to the research?

The necessary inputs have been added

Is the research design appropriate?

The necessary inputs have been added

Are the methods adequately described?

The necessary inputs have been added

Are the results clearly presented?

Thank you for your positive comments

Are the conclusions supported by the results?

Thank you for your positive comments

I thank the authors for their positive response to the most of the comments and suggestions.

Thank you for your positive comments

We expect now the ms is suitable for publication in the Journal.